A pooled analysis of the incidence and mortality risk of atrial fibrillation in patients with COVID-19

Shen Nan-Nan 1
Wang Jia-Liang 1
Liu Xin-Wen 1
Fu Yong-Ping 2 3807944@qq.com
Chen Xue-Fang 3 1522293394@qq.com
1 Department of Pharmacy, Affiliated Hospital of Shaoxing University , Shao Xing, Zhejiang , China
2 Department of Cardiology, Affiliated Hospital of Shaoxing University , Shao Xing, Zhejiang , China
3 Department of Medical Laboratory Science, Affiliated Hospital of Shaoxing University, Shao Xing, Zhejiang Province , Shaoxing , China
Uversky Vladimir
Electronic publication date: 2024 Oct 16
Publication date: 2024
Volume: 12
Electronic Location ID: e18330
Received 2024 Mar 12; Accepted 2024 Sep 25
Copyright: © 2024 Shen et al.
Copyright year: 2024
Copyright holder: Shen et al.
License: This is an open access article distributed under the terms of the Creative Commons Attribution License, which permits unrestricted use, distribution, reproduction and adaptation in any medium and for any purpose provided that it is properly attributed. For attribution, the original author(s), title, publication source (PeerJ) and either DOI or URL of the article must be cited.
License URL: https://creativecommons.org/licenses/by/4.0/

Keywords: Atrial fibrillation, COVID-19, Prevalence, Mortality

Funding: Health Science and Technology Project of Shaoxing 2023SKY079 Clinical Medical Research Special Fund Project of Zhejiang Medical Association 2023ZYC-A55, 2022ZYC-Z37 Program of General Scientific Project of Zhejiang Education Department Y202249053 Research Project of Grassroots health science of Zhejiang Province 2022ZD09 Zhejiang Pharmaceutical Society Hospital Pharmacy Special Research Project 2016ZYY29 This study was supported by the Health Science and Technology Project of Shaoxing (2023SKY079), the Clinical Medical Research Special Fund Project of Zhejiang Medical Association (2023ZYC-A55), the Clinical Medical Research Special Fund Project of Zhejiang Medical Association (2022ZYC-Z37), the Program of General Scientific Project of Zhejiang Education Department (Y202249053), the Research Project of Grassroots health science of Zhejiang Province (2022ZD09), and the Zhejiang Pharmaceutical Society Hospital pharmacy special research project (2016ZYY29). The funders had no role in study design, data collection and analysis, decision to publish, or preparation of the manuscript.

==============================
Background

There exist serious cardiovascular complications subsequent to SARS-Cov2 infection (COVID-19); however, the association between COVID-19 and atrial fibrillation (AF) remains to be elucidated. We aimed to assess the prevalence of AF among COVID-19 patients and its associated risk of death.

Methods

The present systematic review was performed in accordance with the PRISMA guidelines. The protocol was registered with CRD42022306523. A comprehensive literature search was performed across PubMed, Embase, and Cochrane databases to identify studies reporting on the prevalence of pre-existing or new-onset fibrillation (AF), and/or the associated clinical outcomes in patients with COVID-19 from January 2020 to December 2023. The random-effect model was used to estimate the prevalence of AF and its related mortality.

Results

A total of 80 studies, including 39,062,868 COVID-19 patients, were identified in the present investigation. The prevalence rates of pre-existing AF or new-onset AF were 10.5% (95% CI [9.3–11.7%]) or 10.3% (95% CI [6.2–14.5%]), respectively. Subgroup analysis revealed a two fold higher incidence of AF in older patients (≥65 years) compared to younger patients (<65 years) (14.4% vs. 6.4%). The highest rate of AF was observed in Europeans (10.7%, 95% CI [10.2–11.2%]), followed by Northern Americans (10.0%, 95% CI [8.2–11.7%]), while Asians demonstrated a lower prevalence (2.7%, 95% CI [2.2–3.3%]). Notably, severe COVID-19 patients displayed a significantly elevated prevalence of AF at 14.l% (95% CI [13.3–14.9%]), which was approximately 2.5-fold higher than that in non-severe patients (5.2%, 95% CI [4.8–5.5%]). Both pre-existing (HR: 1.83, 95% CI [1.49–2.17]) and new-onset AF (HR: 3.47, 95% CI [2.26–5.33]) were associated with an increased mortality risk among COVID-19 patients. Furthermore, the effect on mortality risk was more significant in Asians (HR: 5.33, 95% CI [1.62–9.04]), compared to Europeans (HR: 1.68, 95% CI [1.24–2.13]) and North Americans (HR: 2.01, 95% CI [1.18–2.83]).

Conclusion

This study comprehensively investigated the association between AF and COVID-19 in a real-world setting. Notably, a high prevalence of AF was observed among older individuals, severe COVID-19 patients, and in Europe and Northern America. Moreover, co-existing AF was found to be associated with an increased risk for mortality. Further investigations are warranted to improve the management and outcomes of COVID-19 patients with AF.

Introduction

The COVID-19 pandemic, starting from in early 2020 and continues to persist, has garnered significant attention due to its high rates of mortality and morbidity (Lotfi, Hamblin & Rezaei, 2020; Zhai et al., 2020; Shereen et al., 2020). In addition to primarily targeting the respiratory system, the original strain of SARS-CoV-2 also attacks multiple organs (Berlin, Gulick & Martinez, 2020). Notably, complications involving the cardiovascular system have emerged as one of most prevalent and severe manifestations during the development of COVID-19 (Wang et al., 2020). These encompass cardiac arrhythmias, myocardial infarction, cardiomyopathy and atrial fibrillation (AF), particularly among individuals with pre-existing conditions.

The prevalence of AF is increasing in the elderly population, leading to an increased rates of morbidity and mortality (Iwasaki et al., 2011). There exists a strong association between AF and clinical outcomes in COVID-19 patients (Sanz et al., 2021; Paris et al., 2021; Fried et al., 2020), particularly in cases of new-onset AF diagnosed after confirmation of COVID-19 without prior history of AF, which subsequently complicates the condition in critically ill individuals (Spinoni et al., 2021). These findings align with previous studies demonstrating a close correlation between the prevalence of AF and its adverse effect on clinical consequences in COVID-19 patients (Mountantonakis et al., 2021; Linschoten et al., 2020). However, there exist controversial publications regarding the varying incidence of pre-existing or new-onset AF and its impact on mortality due to differences in sample size and geographical factors (Mountantonakis et al., 2021; Colon et al., 2020; Rav-Acha et al., 2021). Specifically, certain study has demonstrated a high occurrence of AF and identified it as an independent predictor of mortality (Mountantonakis et al., 2021). Conversely, other studies present contrasting findings (Colon et al., 2020; Rav-Acha et al., 2021). Therefore, it is imperative to conduct comprehensive analyses encompassing diverse regions to elucidate the association between AF and COVID-19.

Previous meta-analyses have documented the association between AF and COVID-19 (Zuin et al., 2021; Li et al., 2021; Romiti et al., 2021; Chen et al., 2021). However, these studies present with relatively a limited number of literature and smaller patients cohorts. Furthermore, these meta-analyses did not specifically focus on AF episodes, or differentiate between different types of arrhythmias, which may compromise the overall conclusion. Moreover, subgroup analyses were not specifically designed to evaluate the prevalence of AF, particularly new-onset AF, and its impact on outcomes in COVID-19. Considering these limitations, there is an urgent need for a systematic reassessment of this topic using rigorous methodology. Therefore, our meta-analysis strictly adhered to predefined inclusion criteria and focused exclusively on episodes of AF.

In the current study, we conducted a comprehensive systematic review and meta-analysis to evaluate the association between AF and COVID-19, aiming to identify the actual incidence of AF and its related outcomes among patients with COVID-19.

Methods

Search strategy

This study was prospectively registered in PROSPERO (ID: CRD42022306523), following the Preferred Reporting Items for Systematic Reviews and Meta-analysis (PRISMA) guideline (Moher et al., 2009). Jia-Liang Wang and Nan-Nan Shen independently conducted comprehensive searches of electronic databases, including Pubmed, Embase, and Cochrane, to identify relevant studies investigating the prevalence of AF, and associated outcomes in COVID-19 patients from inception until December 31, 2023, irrespective of languages. In cases of discrepancies, consensus was achieved with Yong-Ping Fu. The detailed search terms and strategy were outlined in eTable S1. Additionally, all relevant literature from included studies underwent thorough screening to identify potential studies.

Study selection

The inclusion criteria comprised the following: studies involving patients with confirmed diagnosis of COVID-19, reporting data on the prevalence of AF, and/or associated outcomes in COVID-19 patients. The exclusion criteria included sample sizes less than 10, case reports, review articles, conference abstracts, and letters. If the same cohort was reported in multiple studies, the study providing the most comprehensive information was included. Jia-Liang Wang and Nan-Nan Shen systematically screened titles and abstracts to identify potentially relevant full-texts. Any disagreements were resolved through consensus achieved by Yong-Ping Fu.

Data extraction

Two researchers independently extracted data using a pre-designed protocol, including study characteristics, demographics, and concomitant treatment drugs. The data pertaining to the severity of illness in patients were also extracted, severe COVID-19 patients were defined as those requiring admission to the intensive care unit (ICU) or explicitly described as “critical” in the original studies. The prevalence of AF was categorized into pre-existing and new-onset AF. Pre-existing AF was documented history or confirmed diagnosis of AF prior to their COVID-19 diagnosis. New-onset AF was defined as patients who received an AF diagnosis subsequent to confirming COVID-19 infection without any previous history of AF.

Quality assessment

The quality of each included study was independently assessed by two investigators using the modified version of the Newcastle-Ottawa Scale (NOS), which comprised five domains: representativeness of sample population, sample size, participation rate, outcome assessment, and analytical methods to control for bias (Cota et al., 2013). Each item could receive a maximum score of 2 points, resulting in a total score ranging from 0 to 10 points (eTable S2). Studies with a cumulative score ≤ 6 were classified as having a high risk of bias.

Data synthesis and statistical analysis

The random effects model was employed to calculate the pooled prevalence and hazard ratio (HR), and presented with their corresponding 95% confidence intervals (CIs). Heterogeneity among studies was assessed using I2 statistic, with I2 > 50% indicated substantial heterogeneity. Subgroup analyses were conducted to explore potential sources of heterogeneity according to mean age, study type, sample size, severity of illness, and different regions (Europe, North America, and Asia). The comparability in each subgroup was assessed by the interaction analyses (P for interaction). Sensitivity analysis was carried out by systematically excluding individual studies to evaluate the robustness and reliability of the primary analysis. Meta-regression analysis was performed to assess the potential impact of baseline cofounders (mean age, female ratio, HF, HBP, DM, BMI) on the pooled results. Publication bias was qualitatively assessed using funnel plots and quantitatively evaluated through Begg’s test and Egger’s test (Egger et al., 1997). All the statistical analyses were performed using STATA version 13.0 (Statacorp, CollegeStation, Texas, United States).

Results

Study search and selection

The flow diagram illustrating the literature retrieval process is presented in Fig. 1. A total of 1,825 relevant records were obtained from electronic databases, including Pubmed (1,439), Embase (261), and Cochrane (125), using the specified search terms. After removing duplicates, 1,736 records remained for further analysis. Following screening of titles and abstracts, a total of 133 full-text articles were selected for inclusion in the current study. Subsequently, we excluded additional 53 articles based on the following criteria: No prevalence data (n = 35); Review without data (n = 12); Small sample size (n = 1); Arrhythmias without AF (n = 1); AF/atrial flutter (n = 4) (eTable S3). Finally, a total of 80 studies involving 39,062,868 COVID-19 patients were finally included in this study.

Figure 1 Flow diagram for the selection of eligible studies.

AF, atrial fibrillation.

Study characteristics

The main characteristics of the 80 articles are summarized in Table S1. Among the included studies, 15 studies were performed in Asia (four in China, six in Turkey, two in India, two in Korea and one in Saudi Arabia), while North America accounted for 22 studies (all conducted in the USA). Europe contributed to the highest number of studies with a total of 43 (19 from Italy, seven from UK, seven from Spain, four from Denmark, and one each from France, Netherlands, Poland Portugal and Switzerland). Out of all included studies, retrospective designs were employed by seventy-one studies while nine utilized prospective approaches. Seventy-four studies reported the prevalence of pre-existing AF, three studies reported the prevalence of new-onset AF, additionally three other studies provided data on both pre-existing and new-onset prevalence. Furthermore, twenty-eight articles reported data on all-cause mortality. The sample size of the included studies ranged from 30 to 25,333,329, with a total of 39,062,868 individuals (Table S1).

Patient characteristics and quality assessment

The detailed patient characteristics were shown in eTable S4. The mean age of patients ranged from 44.8 to 86.3, with 41.2% being female. Hypertension (50.7%), Diabetes Mellitus (25.1%), and heart failure (11.4%) were the most common comorbidities of cardiovascular diseases observed among the patients included in this study. Quality scores for all included studies are reported in eTable S5, indicating moderate to high quality assessments ranging from a score of 6 to 9.

Pooled prevalence of AF in patients with COVID-19

The overall and subgroup prevalence of AF were outlined in Fig. 2. The estimated global prevalence of pre-existing AF was 10.5% (95% CI [9.3–11.7%], I2: 100%) (eFig. S1). In subgroup analysis of pre-existing AF, older patients demonstrated approximately a 2.5-fold higher prevalence (14.4%, 95% CI [12.2–15.7%], I2: 99.8%) compared to younger patients (6.4%, 95% CI [5.6–7.3%], I2: 97.4%) (eFigs. S2 and S3). Furthermore, significant differences were observed among different geographic regions, with the highest prevalence observed in Europeans (10.7%, 95% CI [10.2–11.2%], I2: 99.9%), followed by Northern Americans (10.0%, 95% CI [8.2–11.7%], I2: 99.8%), Asians (2.7%, 95% CI [2.2–3.3%], I2: 89.2%) (eFigs. S4–S6). The prevalence of AF in severe COVID-19 patients (14.1%, 95% CI [13.3–14.9%], I2: 98.4%) was approximately 2.5-fold higher than non-severe patients (5.2%, 95% CI [4.8–5.5%], I2: 92.7%) (eFigs. S7 and S8). The subgroup of sample size showed that the incidence rate was about 1.3-fold higher for small sample size (12.5%, 95% CI [10.0–15.1%], I2: 39.6%) compared to large sample size (9.7%, 95% CI [8.2–11.2%], I2: 99.9%) (eFigs. S9 and S10).

Figure 2 Pooled prevalence of AF in COVID-19.

No., number of included studies; AF, atrial fibrillation.

Due to the relatively small number of new-onset AF, the subgroup analyses based on age stratification, sample size, and study design were not feasible. Consistent with the overall prevalence (10.3%, 95% CI [6.2–14.5%], I2: 93.6%), the prevalence in patients with severe and non-severe AF patients were 12.2% and 6.6%, respectively (eFig. S11). However, this data is not statistically significant and should be interpreted cautiously.

The effect of co-existing AF on all-cause mortality in patients with COVID-19

A total of 25 articles, involving 31,471,171 COVID-19 patient, reported the association between pre-existing AF and the risk of all-cause mortality in COVID-19 patients. Pre-existing AF (HR: 1.83, 95% CI [1.49–2.17], I2: 92.3%) was significantly associated with an increased risk of mortality in COVID-19 patients (Fig. 3, eFig. S12). Subgroup analysis by geographic region revealed a approximately 3.0-fold higher risk of mortality in Asians (HR: 5.33, 95% CI [1.62–9.04], I2: 56.8%) compared to Europeans (HR: 1.68, 95% CI [1.24–2.13], I2: 89.1%) and North Americans (HR: 2.01, 95% CI [1.18–2.83], I2: 94.8%) (eFig. S13). However, the caution should be exercised in interpreting this finding due to the statistical P-value exceeding the threshold of significance (P > 0.05). The risk for mortality was found to be significantly higher in retrospective studies (HR: 1.95, 95% CI [1.52–2.39], I2: 93.1%) compared to prospective studies (HR: 1.26, 95% CI [0.69–1.83], I2: 15.4%) (eFig. S14). The studies involving more than 500 individuals revealed a significantly higher death risk (HR: 2.04, 95% CI [1.62–2.46], I2: 95.9%). In addition, the death risk did not show a statistical difference in the subgroup of age (elderly vs. young: 2.13 vs. 1.90), severity of illness (severe vs. non-severe: 1.91 vs. 1.51) (eFigs. S15–S17).

Figure 3 Risk of all-cause mortality in AF vs. non-AF patients.

No., number of included studies; AF, atrial fibrillation.

Four studies involving 1,034 individuals examined the association between mortality and new-onset AF. Compared to pre-existing AF, new-onset AF was significantly associated with a 1.8-fold higher risk of death in COVID-19 patients (OR: 3.47, 95% CI [2.26–5.33], I2: 0.0%, P < 0.001) (eFig. S18). However, due to the limited number of available studies, further subgroup analysis could not be conducted.

Sensitivity analysis and meta-regression

Sensitivity analyses were conducted to explore the robustness of primacy results. Each study was sequentially removed to evaluate its impact on the pooled estimates. The results of sensitivity analysis suggested that none of the studies significantly influenced the overall prevalence and mortality rates (eTables S6–S8). Meta-regression analysis was performed to investigate potential patient characteristics affecting both AF prevalence and associated mortality in COVID-19 patients. The findings revealed a significant association between mean age and AF prevalence in individuals with COVID-19 (P < 0.05). Conversely, no significant correlations were observed between other variables and either pooled prevalence or mortality rates (eTables S9 and S10). Sensitivity analysis further confirmed the robustness of these results, as no individual study caused a reversal of the pooled effect size.

Publication bias

The risk of publication bias was assessed by Begg’s test and Egger’s regression test (eFigs. S19 and S20). No significant evidence of publication bias was observed in pooled prevalence of pre-existing AF (P for Begg’s test: 0.839; P for Egger’s test: 0.775), as well as in mortality (P for Begg’s test: 0.413; P for Egger’s test: 0.399). Due to the limited number, the funnel plot analysis was not performed for new-onset AF.

Discussion

Overall, the 80 studies were identified, involving a substantial cohort of 39,062,868 COVID-19 patients. Our findings demonstrated that the prevalence of AF in COVID-19 surpasses 10%, particularly among elderly patients and those from Europeans and North Americans with severe COVID-19 infection. Furthermore, our investigation reveals a significant association between pre-existing AF and an increased risk of mortality in COVID-19 patients, especially among Asians or individuals developing new-onset AF.

In the present studies, a comprehensive search was conducted using stricter inclusion criteria and a rigorous methodology approach to obtain more reliable and accurate data. Our findings demonstrated thatthe incidence of atrial fibrillation (AF) among COVID-19 patients was 10%, which is in line with previous study (Cangemi et al., 2015), suggesting that SARS-Cov2 virus also attacks hearts, in addition to the respiratory system during COVID-19 development, possibly through interaction with the S protein on target cells. Previous studies have shown a correlation between AF and SARS-CoV-2 infection in both prospective and retrospective studies (Velilla-Alonso et al., 2021; Terlecki et al., 2021; Phelps et al., 2021). However, our study expands on these findings by revealing a high prevalence of AF among older age groups and/or severe COVID-19 patients, particularly in Europeans and North Americans. The strength of our study lies in its extensive inclusion of numerous articles encompassing a significantly larger patient population size compared to previous studies (Li et al., 2021; Zhou et al., 2021; Romiti, 2021). Consequently, our findings further enhance the generalizability of these results.

We observed a significant increase in mortality when atrial fibrillation (AF) coincided with COVID-19, particularly in cases of new-onset AF. This may be attributed to the higher incidence of embolism events associated with new-onset AF, leading to worse cardiovascular outcomes (Sanz et al., 2021). Despite Asians having the lowest occurrence of AF and COVID-19 infection, they exhibited the highest mortality following SARS-Cov2 infection according to subgroup analysis based on different geographic regions. We hypothesize that this increased mortality among Asians could be due to more severe illness and older age among COVID-19 patients (Özdemir, Özlek & Çetin, 2021; Mathew et al., 2021; Fumagalli et al., 2021). However, further investigation is required to elucidate the precise underlying mechanism of AF prevalence among COVID-19 patients, including multi-centers and different racial backgrounds for both improving the clinical outcomes and informing policy-making.

Currently, the pathogenesis of COVID-19 related AF remains poorly understood. It is suggested that SARS-CoV-2 virus is targets cells via ACE2 receptor (Kochi et al., 2020), which is abundant in respiratory and cardiovascular systems (Tortorici et al., 2019). Specifically, the virus invades various host cells, including pneumocytes, endothelial cells, and cardiomyocytes via ACE2. Consequently, a large amount of release of angiotensin II (AngII) from COVID-19 patients may contribute to AF among susceptible individuals (South, Diz & Chappell, 2020; McKinney et al., 2014). This hypothesis is supported by studies indicating that AF is an independent risk factor for worse prognosis and increased mortality in COVID-19 patients (Iaccarino et al., 2020; Mair, Foster & Nicholl, 2020). Although there is currently no definitive evidence linking AF to increased death in these patients, autopsy findings have revealed a substantial infiltration of macrophages into myocardial tissues of COVID-19 cases (Basso et al., 2020). Additionally, COVID-19 plays a major role in coagulopathy leading to microvascular and macrovascular thrombosis that can cause cardiac injury along with distant embolism (Ackermann et al., 2020; Patel et al., 2020). Such speculation is consistent with previous large cohort studies demonstrating high thromboembolic risk among infected individuals with AF (Gundlund et al., 2020; Hindricks et al., 2021), thus emphasizing the necessity for anticoagulant therapy in such cases. Therefore, this study provides valuable insights into the management of concurrent atrial fibrillation in COVID-19 patients, aligning with European guidelines on atrial fibrillation to enhance prognosis.

Study limitations

Several limitations exist in our study. Firstly, real-world observational studies are inherently biased and exhibit a high degree of heterogeneity, despite the utilization of the random effects model for estimating pooled analyses of prevalence and mortality risk. Secondly, the adjustment for confounding factors on mortality was performed in limited number of studies, potentially leading to bias primarily due to the nature of real-world research. Thirdly, subgroup analysis regarding new-onset AF could not be conducted due to insufficient data, however, this aspect will be addressed in future investigations. Further investigation is warranted to explore safe and effective strategies for clinical treatment and management of COVID-19 patients with AF.

Conclusion

Our findings indicate a significantly elevated prevalence of AF in COVID-19 patients, which is associated with an increased risk of mortality, particularly among older individuals and those with severe COVID-19. Therefore, it is imperative to prioritize the management of AF in COVID-19 patients to enhance their clinical outcomes.

Supplemental Information

Supplemental Information 1 PRISMA 2020 checklist.

Supplemental Information 2 Supplementary Materials.

Additional Information and Declarations

Competing Interests

Author Contributions

Data Availability

The authors declare that they have no competing interests.

Nan-Nan Shen conceived and designed the experiments, analyzed the data, prepared figures and/or tables, and approved the final draft.

Jia-Liang Wang conceived and designed the experiments, authored or reviewed drafts of the article, and approved the final draft.

Xin-Wen Liu analyzed the data, prepared figures and/or tables, and approved the final draft.

Yong-Ping Fu conceived and designed the experiments, authored or reviewed drafts of the article, and approved the final draft.

Xue-Fang Chen performed the experiments, analyzed the data, prepared figures and/or tables, and approved the final draft.

The following information was supplied regarding data availability:

The raw measurements are available in the Supplemental File.

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
