# Peer review of "A pooled analysis of the incidence and mortality risk of atrial fibrillation in patients with COVID-19"

_PeerJ, doi:10.7717/peerj.18330_

## Round 0.1 · original submission · Minor Revisions

Please address all the issues pointed by the reviewers and amend the manuscript accordingly.

Reviewer 1 ·

Basic reporting

Please review the text to remove any grammatical errors and ensure a smooth sentence-to-sentence and paragraph-to-paragraph transition.

The figure quality is not great. Their resolution can be improved.

Experimental design

Project objectives are well explained. Experimental design is well-defined and relevant.

Validity of the findings

Data and findings are robust and relevant conclusions are drawn from the data.

Reviewer 2 ·

Basic reporting

1) Language is clear.

Experimental design

2) A reference number has to be indicated for each pointed study in Table 1.
3) The country column has a typo, 'Clift', in Table 1.
4) Authors should indicate the number of patients with AF also in Table 1 near the column to the total number of patients with Covid 19.

Validity of the findings

The conclusions are well-written and linked to the aim of the research.

Reviewer 3 ·

Basic reporting

The manuscript titled “A pooled analysis of the incidence and mortality risk of AF in patients with COVID-19” by Shen et al. shows the prevalence and health risks of AF in COVID-19 patients. The authors should be commended for rigorous analysis of previously published results. However, several issues beg attention, and I have listed them here-

1. In certain areas of the manuscript, the English language should be improved to ensure that an international audience can clearly understand your text. Some examples where the language could be improved include lines 73, 195-196, 221, 235-239 – the current phrasing makes comprehension difficult. I suggest you have a colleague proficient in English and familiar with the subject matter review your manuscript or contact a professional editing service.
2. In lines 62-63, the authors need to provide more detail about the controversial publications, for example, the reason for the controversy, so that the readers can understand the necessity of the present study.
3. The authors need to explain the meaning of new-onset AF in the introduction.
4. Minor typographical errors that are present throughout the manuscript should be corrected.
5. Each of the subsections in the results should begin with a statement about what the authors gain from the analysis and end with a summary of an explanation of the findings from that section.
6. References need to be included in appropriate places. For example, line 228 can benefit from a reference to the previous studies.

Experimental design

The experiments were well designed.

Validity of the findings

The validity of some of the findings is questionable, and they are enumerated below.
1. Among the new-onset AF data, the P value and the error bars of the severity of illness, seems significantly high. Any analysis based on the data will be prone to errors, so the data should be removed.
2. The error bar for the Asian group in Fig. 3 seems to be higher compared to the other two, and the P value is also significantly greater than 0.05. Therefore, based on the results, the inference on lines 180-184 is incorrect.
3. There is a negligible difference in the death risk observed between patients above and below 65 years of age and in severe and non-severe patients in Fig. 3. The inference based on these results is incorrect.

---

## Round 0.2 · accepted · Accept

All issues of the reviewers were addressed and revised manuscript is acceptable now.

Reviewer 3 ·

Basic reporting

The authors have appropriately addressed all my concerns. I am satisfied with the disclaimer sentences added by the authors for the statistically insignificant data.

Experimental design

No comment

Validity of the findings

No comment